# Alternative Splicing and Alternative Polyadenylation-Regulated Cold Stress Response of *Apis cerana*

**DOI:** 10.3390/insects15121006

**Published:** 2024-12-19

**Authors:** Yuanchan Fan, Dan Yao, Jinmeng Ma, Fangdong You, Xiaoping Wei, Ting Ji

**Affiliations:** 1Apicultural Research Institute, College of Animal Science and Technology, Yangzhou University, Yangzhou 225009, China; yuanchanfan1997@163.com (Y.F.); m19505501535@163.com (J.M.); 2Guizhou Institute of Integrated Agriculture Development, Guizhou Academy of Agricultural Sciences, Guiyang 550006, China; dyao20241218@126.com; 3Yunnan Provincial Department of Agriculture and Rural Affairs, Yunnan Animal Husbandry Station, Kunming 650225, China; apiary@163.com

**Keywords:** honeybee, *Apis cerana*, alternative splicing, alternative polyadenylation, cold stress, cold tolerance

## Abstract

Bees, while individually poikilothermic, can maintain group warmth, which is vital for biodiversity. *Apis cerana* outperforms *Apis mellifera* in cold tolerance, particularly as a pollinator in harsh conditions. Here, we utilized transcriptome data to identify alternative splicing events and alternative polyadenylation genes associated with cold stress in *A. cerana*. These genetic changes are linked to signal transduction, energy metabolism, and oxidative stress, with key genes like *HSP70* and *BAG* increasing in expression and others showing a pattern of an initial decrease followed by an increase. This post-transcriptional regulation may help bees adapt to cold, providing insights for breeding cold-resistant bee strains and advancing cold resistance research in insects.

## 1. Introduction

Temperature is a pivotal ecological factor in the regulation of insect survival and reproduction. In temperate regions, where temperature fluctuations are common, insects face continuous survival challenges due to temperature stress. The ability to withstand low temperatures is essential for insect population survival and reproduction. [1]. The low-temperature test brought by the long winter is the premise of its population survival and reproduction. Insects regulate behaviors such as migration; rhythm regulation; physiological regulation, such as increasing the expression of antifreeze proteins [2], heat shock proteins [3], glycerol, sugar, and other cryoprotectants, regulating metabolic rate and water content, diapause [4], and increasing polyol accumulation [5]; and adaptive changes in external body surface structure, such as changing body color [6], exoskeleton thickness, hair and scale density, and other strategies to cope with low-temperature stress [7]. At present, the cold tolerance of insects is mostly concentrated in solitary insects, and the individual temperature regulation of social insects is relatively lower [8]. Honeybees, as true social insects, play a critical role in plant pollination and reproduction, making the understanding of their cold tolerance mechanisms imperative. *Apis cerana* is a native bee species in East Asia. It has stronger cold tolerance than *Apis mellifera* and is a dominant sample for cold tolerance research [9]. Worker bees with different roles face distinct temperature challenges and employ various coping strategies. For foraging bees outside of the nest, the optimal temperature for collection flights is between 18 °C and 30 °C. The critical temperature for individual safety for *A. cerana* is 10 °C, and, below 6.5 °C, they cease activities outside of the nest. Foraging bees increase their body temperature by consuming glycogen substances to prevent their wing muscles from freezing, ensuring their ability to fly [10,11,12]. For nurse bees inside the nest, when the ambient temperature ranges from −1.91 °C to 7.62 °C, the center temperature of the colony is maintained between 19.75 °C and 26.44 °C, while the edge temperature is between 12.91 °C and 14.60 °C [13]. To maintain the nest temperature, the nurse bees keep the central nest area warm through clustering together. However, current research on their low-temperature tolerance remains quite limited, and comprehensive data to systematically explain advancements in this field are still lacking.

In recent years, a plethora of studies have demonstrated that the response of organisms to temperature stress is related to post-transcriptional regulation. The post-transcriptional regulatory mechanisms involved in the epigenetic forms of gene regulation in eukaryotic organisms include alternative splicing (AS) and alternative polyadenylation (APA). AS is a post-transcriptional regulatory mechanism that allows for the generation of multiple unique mRNAs from the untranslated region (UTR) or coding sequence (CDS) of a single gene. The regulation mode of AS is constantly changing under physiological conditions, enabling organisms to respond to environmental changes by determining which specific parts of the genome they express [14]. AS can be categorized into the following seven types: Skipped exon (SE), Mutually exclusive exon (MX), Alternative 5′ splice site (A5), Alternative 3′ splice site (A3), Alternative first exon (AF), Alternative last exon (AL), and Retained intron (RI). AS widely regulates insect life activities, such as growth and development, response to environmental stress, genetic differentiation, habits, and behaviors. For instance, in *Drosophila*, heat shock transcription factor 1 (HSF1) regulates the expression of heat shock proteins through AS, producing different subtypes of *DHSFB*, *DHSFC* and *DHSFD*. *DHSFB* is highly expressed under heat stress and *DHSFD* is highly expressed under cold stress. AS plays a crucial role in the temporal and spatial expression of heat shock proteins [15]. Significant differences in AS can be observed among various genotypes of *Drosophila* exposed to extreme temperatures. Similar to the differences between species genes, genes with differential AS between different genotypes of *Drosophila* usually show dominant inheritance, and the AS changes caused by temperature may be heritable [16]. APA further regulates protein expression on the basis of AS, affecting the coding potential or 3′ UTR length by altering the binding availability of miRNA or RNA, thereby regulating protein diversity [17]. It is ubiquitous across all eukaryotic species and is regarded as a principal mechanism for gene regulation. It exhibits tissue specificity and plays a crucial role in cell proliferation and differentiation. The identification of APA events using second-generation sequencing technology necessitates specific amplification of the 3′ end, which has resulted in relatively limited research on insects [17]. With the development and application of third-generation sequencing, a multitude of APA events have been identified in the genomes of species such as humans [18], bamboo [19], and sorghum [20]. It has been confirmed that APA may mediate insect cell signal transduction, immune activation, and development [21,22]. Research on desiccation stress in *Polypedilum vanderplanki* has demonstrated that the organism can regulate the expression of heat shock proteins via APA to adapt to dehydration stress [23].

Third-generation sequencing (TGS) technology provides the possibility for high-throughput identification of variable shear events and variable polyadenylation sites. In recent years, measured DNA sequence data have also increased rapidly with the continuous development of biological genome sequencing technology, which has promoted the in-depth development of bioinformatics-related research. First-generation sequencing (FGS) usually refers to Sanger sequencing, which has been applied and promoted in the human genome project. In 2001, Sanger sequencing was used to complete the construction of the human genome block diagram. Subsequently, this technology was applied to the genomics research of *Bombyx mori*, *Drosophila*, and *A. mellifera*. However, due to its high accuracy, low throughput, low coverage, and long sequencing cycle, and it is difficult to achieve such a wide range of applications. Next-generation sequencing (NGS) has the advantages of high throughput and low cost, and it plays an important role in the research of human diseases, animals, plants, and microorganisms. However, its read length is shorter, and it has a GC bias, requiring algorithm optimization to complete the sorting. Additionally, it is unable to accurately detect fused genes and gene families. The TGS represented by PacBio SMRT technology and Oxford Nanopore single molecule technology solves the problems of reading length and template-based amplification associated with FGS and NGS, significantly reducing the amount of subsequent gene splicing; improves the quality of functional annotation; and eliminates the wrong base introduced due to PCR amplification [24]. Compared with genome sequencing, the costs and cycle of transcriptome sequencing using TGS technology are lower, which can improve the accuracy of the data. In the current landscape of insect genomics, TGS has become instrumental in capturing precise shear isomers during periods of physiological stress in insects. This advanced approach enables a thorough comprehension of gene function and regulation, offering insights into the molecular mechanisms that underpin insect stress responses. Through TGS of silkworms, five variable splicing genes have been identified as key regulators in the stress response to *B. mori* nucleopolyhedrovirus [25]. These findings are crucial for the development of innovative control strategies aimed at managing viral infections in silkworm populations. TGS revealed that AS- and APA-generated c-Jun N-terminal kinase (JNK) activation via distinct shear isomers in small brown planthoppers modulates resistance and virus–insect-vector interactions during rice stripe virus infection [26].

In view of the lack of research on the molecular regulation mechanism of cold tolerance in honeybees, here, we used PacBio sequencing technology to perform transcriptome sequencing of *A. cerana* after cold coma and cold coma recovery after 4 °C low-temperature treatment. Transcriptome identification of multiple AS and APA produced a comprehensive post-transcriptional regulatory network map. We annotated the differentially expressed ASs (DASs) and APAs (DAPAs), compared the highly expressed AS and APA regulatory genes by comparing the second-generation transcriptome data, and then studied the pathways and genes related to cold stress and cold tolerance in *A. cerana*. Finally, these key regulatory AS genes and APA genes were verified with PCR and 3′RACE, which will facilitate future research towards elucidating the mechanisms underlying the cold stress response in honeybees.

## 2. Materials and Methods

### 2.1. Honeybees

*A. cerana* workers were selected from three colonies located in the Guizhou institute of modern agricultural development. After marking the newly emerged bees (*Ac*0d) with color and returning them to the hive, they were recaptured once they had reached the appropriate age. The 3- (*Ac*3d), 10- (*Ac*10d), and 21-day-old (*Ac*21d) *A. cerana* workers (*n* = 30) were put into a constant temperature and humidity box with a normal temperature (25.0 ± 0.2 °C) (CK) and a low temperature (4.0 ± 0.2 °C) (relative humidity 60%) for 6 h (T), and then the bees were placed into normal temperature recovery (25.0 ± 0.2) until normal activity (TR). For Illumina RNA-seq second-generation sequencing, 10 bees were used per sample with 3 biological replicates. For PacBio Sequel third-generation sequencing, 15 bees were used per sample. Total RNA was extracted using a TRizol Regent kit (Thermo Fisher, Waltham, MA, USA).

### 2.2. Transcriptome Data Source

The samples of CK, T, and TR groups of the 3 d, 10 d and 21 d *A. cerana* workers were sequenced on PacBio Iso-Seq and Illumina RNA-Seq platforms, respectively. The SMRTlink software (SMRT Link version 13.1) was used to process the original offline data of PacBio, and then the second-generation high-quality data were used to correct the sequencing errors of the third-generation reads. The GMAP (genetic mapping and alignment program) software (GMAP software version 2011-10-16) [27] was used to align the purified consensus sequence to the reference genome (gca_029169275.1). The results show that 310,214 to 959,759 raw reads were generated, and the polymerase read N50 was between 169,861 and 218,264 bp, while the average read length was between 98,397 and 141,959 bp, which suggested that the high-quality, full-length transcriptome data could be used in this study (Appendix A).

### 2.3. Identification of Differential AS

The types of AS mainly include Skipped exon (SE), Mutually exclusive exon (MX), Alternative 5′ splice site (A5), Alternative 3′ splice site (A3), Alternative first exon (AF), Alternative last exon (AL), and Retained intron (RI). The AS type of each group of sample genes was identified with SUPPA software (SUPPA version 2.4) [28], and the default parameters were used. The AS event types of genes were counted according to the identification results. The AS events were highly spatiotemporally specific in their regulation. Consequently, the AS events of *A. cerana* at different age stages and under various temperature treatments exhibited differentiation. Since AS analysis pertains to structural analysis and cannot be quantitatively analyzed, the Blast tool was employed to compare the differential transcripts of *A. cerana* in cold treatment isoforms. DASs were screened according to the criteria of *p* value ≤ 0.05 and |log2 (fold change)| ≥ 1, from the comparisons T vs. CK (cold stress) and TR vs. CK (cold tolerance) at the time points of 3 days, 10 days, and 21 days post-treatment for *A. cerana*.

### 2.4. Identification of Differential APA

The APA loci of genes were identified using the Tapis pipeline [20], employing the default parameters. MEME software (MEME software version 5.3.3) was used to analyze the sequence characteristics 50 bp upstream of the poly(A) shear site of the transcript to identify the motif. Due to APA being a qualitative structure analysis tool, it was impossible to quantitatively screen the differentially expressed genes among the different samples, and it needs to be combined with differential expression gene (DEG) analysis for comparative characterization. Therefore, the Blast tool was used to compare the DAPAs in the cold stress and cold tolerance groups.

### 2.5. GO Enrichment and KEGG Annotation Analysis

GO (Gene Ontology) categorization of isoforms was carried out using WEGO software (WEGO 2.0) [29]. The Blastall tool was employed to conduct pathway analysis by comparing isoforms against the KEGG (Kyoto Encyclopedia of Genes and Genomes) database (https://www.kegg.jp/, accessed on 18 October 2024). Then, a chart was drawn by using the relevant tools of the OmicShare platform (https://www.omicshare.com/tools/, accessed on 21 October 2024).

### 2.6. Investigation of Chilling Stress Response Factor-Associated DASs and DAPAs

In accordance with relevant documents on the temperature stress response of bees and our previous studies on *A. cerana*, ceramide glucosyltransferase, succinate dehydrogenase, cytochrome C, cubilin homolog, purine nucleoside phosphorylase, isocitrate dehydrogenase, glycerol-3-phosphate dehydrogenase, heat shock 70 kDa protein cognate 4, ribosomal protein S6 kinase alpha-3, vitellogenin-6, elongation of very-long-chain fatty acids protein, and Bcl-2 associated athanogene (BAG) -domain-containing protein. Expression clustering analysis of the aforementioned DASs and DAPAs was performed utilizing the OmicShare platform (https://www.omicshare.com/, accessed on 9 October 2024).

### 2.7. PCR Validation

According to the corresponding nucleic acid sequence, specific primers were designed with primer5 software (Appendix A). The total RNA was extracted with an RNA Extraction Kit (TaKaRa, Dalian, China), and the cDNA template was obtained with a PrimeScript™ RT reagent Kit (TaKaRa, Dalian, China) for PCR amplification. The reaction system included the following: 5× TransStart^®^ FastPfu Buffer 10 μL, 2.5 mM dNTPs 4 μL, TransStart^®^ FastPf DNA Polymerase 1 μL, Sterile water 32 μL, cDNA template 1 μL, upstream primer 1 μL, and downstream primer 1 μL. The reaction procedure was conducted at 95 °C for 2 min, followed by 35 cycles at 95 °C for 20 s, 45 °C for 20 s, 72 °C for 1 min, and 72 °C for 5 min. PCR products were observed and photographed with a nucleic acid gel imager after 1.5% agarose gel electrophoresis.

### 2.8. 3′ RACE

According to the corresponding nucleic acid sequence, specific primers were designed with primer5 software (Appendix A). The total RNA was extracted with an RNA Extraction Kit (TaKaRa, Dalian, China), the 3′-Full RACE Core Set with PrimeScript™ RTase (TaKaRa, Dalian, China), and TaKaRa LA Taq^®^ DNA Polymerase (TaKaRa, Dalian, China) was used for 3′RACE. First, Poly(A)+ RNA was reverse transcribed into cDNA using PrimeScript Reverse Transcriptase. The reaction conditions were as follows: 42 °C 60 min, 70 °C 15 min, the nested PCR reaction was carried out with TaKaRa LA Taq, and the reaction conditions were as follows: 42 °C for 60 min, followed by 40 cycles at 70 °C for 15 min, and 59 °C for 30 s, with melting curve program default system settings. The specific process and system of each step were as follows: outer primer PCR reaction, 4 °C for 3 min, followed by 30 cycles at 94 °C for 30 s, 55 °C for 30 s, 72 °C for 1 min, then 72 °C 10 min; then, 1.5% agarose gel was prepared and 20 μL outer primer PCR reaction solution was added into the well for electrophoresis. The nucleic acid gel imager was used for observation and photography. If the fragment band was not ideal, inner primer PCR reaction was performed (inner primer PCR reaction: 94 °C 3 min, followed by 30 cycles at 94 °C for 30 s, 55 °C for 30 s, 72 °C for 1 min, and 72 °C for 10 min). Then, 20 μL inner primer PCR reaction solution was added into 1.5% agarose gel wells for electrophoresis. The nucleic acid gel imager was used for observation and photography.

## 3. Results

### 3.1. AS Analysis

In total, 25,443 AS events were identified from nine samples without transcriptome assembly, and a total of 2711 genes underwent AS. The full-length transcriptome data identified seven types of AS, as follows: 4636 SE, 493 MX, 2743 RI, 4,655 A5, 4864 A3, 4901 AF, and 619 AL (Figure 1, Appendix A). These AF events account for the majority of AS events, and MX was the least frequent event in AS. Through the classification and analysis of AS in the different-day-old treatment groups, we found that AS events in bees showed significant differences in the cold treatment and cold treatment recovery stages. The AS events in the CK group were more common than those in the T group, while the opposite trend was observed in the T group compared with the TR group. The trend of AS event frequency change was the same in the three developing stages. The *Ac*21d treatment group showed more complex splicing, indicating that AS may be involved in the regulation of 3′ end processing in the cold stress response regulation process of *A. cerana*, which is more frequent in the foragers.

### 3.2. Identification of DASs

In total, 48 DASs were identified in CK vs. T and CK vs. TR at 3 d, 10 d, and 21 d *A. cerana* workers. GO term analysis suggested that the DASs were enriched in 541 terms, and the binding, cellular process, metabolic process, protein binding, and organic substance metabolic process were significantly engaged (Figure 2A). In addition, the KEGG pathway analysis indicated that eight DASs relevant to 14 pathways were present, which were nicotinate and nicotinamide metabolism (ame00760), 2-Oxocarboxylic acid metabolism (ame01210), purine metabolism (ame00230), protein processing in endoplasmic reticulum (ame04141), citrate cycle (TCA cycle) (ame00020), mTOR signaling pathway (ame04150), biosynthesis of amino acids (ame01230), glycerophospholipid metabolism (ame00564), pyrimidine metabolism (ame00240), carbon metabolism (ame01200), endocytosis (ame04144), oxidative phosphorylation (ame00190), spliceosome (ame03040), and metabolic pathways (ame01100) (Figure 2B).

### 3.3. Chilling Stress Response Factor-Associated DASs in A. cerana

The corresponding DASs involved in cold stress in each comparison group were further analyzed. In total, 10 and 42 DASs were identified in cold stress (T vs. CK) and cold tolerance processes (TR vs. CK), respectivley. Additionally, Venn analysis showed that there were four DASs in the two aforementioned processes (Figure 3A). Further analysis of the functional regulation of DASs revealed a total of eight chilling stress response factors, including purine nucleoside phosphorylase (*PNP*), isocitrate dehydrogenase (*IDH3G*), glycerol-3-phosphate dehydrogenase (*GPDH*), heat shock 70 kDa protein cognate 4 (*HSP70*), ribosomal protein S6 kinase alpha-3 (*RPS6KA3*), vitellogenin-6 (*VIT-6*), elongation of very-long-chain fatty acids protein (*AAEL008004*), BAG-domain-containing protein (*SAMUI*) (Figure 3B). Moreover, the expression clustering showed that *HSP70* were induced to activation, and *VIT-6* and *AAEL008004* were suppressed. Trend analysis shows that the expression levels of these cold tolerance and cold stress regulatory genes significantly increase with the recovery time of bees from a chilling coma (*p* < 0.01) (Figure 3C).

### 3.4. APA Analysis

A total of 2821 genes were analyzed for polyadenylation sites, and the distribution of APA site numbers also varies among different treatment groups. Further analysis revealed that the same gene has different numbers of APA sites in the four sample groups (Figure 4A). In addition, there was no significant deviation in nucleotide composition between the upstream and downstream regions of the full-length transcript 3′ UTR. In the upstream region of APA sites in the full-length transcripts of the Eastern honeybee, 20 motifs were identified, with the most frequent being AAUAAA (Figure 4B). The proportion of poly(A) sites > 5 in the TR group was higher than that found in the other two groups, indicating that the 3′ end complexity caused by APA may play an important role in the repair of cold damage in honeybees.

### 3.5. Function and Pathway Annotation of DAPAs

From the cold stress (T vs. CK) and cold tolerance process (TR vs. CK) comparison groups for the 3 d, 10 d and 21 d *A. cerana* workers, we selected genes that exhibit differential expression and undergo APA for Gene Ontology (GO) annotation. The enrichment analysis revealed that DAPAs were significantly associated with the binding, metabolic process, cellular process, catalytic activity, cell, and cell parts (Figure 5A). According to the KEGG pathway annotation results, DAPAs were associated with a total of 16 pathways, including the following: protein processing in endoplasmic reticulum (ame04141), citrate cycle (TCA cycle) (ame00020), nicotinate and nicotinamide metabolism (ame00760), sulfur metabolism (ame00920), 2-Oxocarboxylic acid metabolism (ame01210), carbon metabolism (ame01200), oxidative phosphorylation (ame00190), sphingolipid metabolism (ame00600), spliceosome (ame03040), metabolic pathways (ame01100), biosynthesis of amino acids (ame01230), glycerophospholipid metabolism (ame00564), RNA degradation (ame03018), pyrimidine metabolism (ame00240), endocytosis (ame04144), and purine metabolism (ame00230) (Figure 5B). These metabolic pathways are mainly involved in energy metabolism and oxygen metabolism, indicating that a large number of genes regulate energy metabolism and oxidative stress through APA during cold stress response.

### 3.6. Chilling Stress Response Factor-Associated DAPAs in A. cerana

In total, 12 and 53 DAPAs were identified in cold stress (T vs. CK) and cold tolerance processes (TR vs. CK), respectivley. Venn analysis revealed that there were eight DAPAs in the aforementioned two processes (Figure 6A). Then, a total of four chilling stress response factors were identified in these two processes, which encompass ceramide glucosyltransferase (*UGCG*), succinate dehydrogenase (*SDHB*), cytochrome c (*CYTC*), and cubilin homolog (*CUBN*) (Figure 6B). Trend analysis shows that the expression levels of these cold tolerance and cold stress regulatory genes significantly decreased first and then increased (*p* < 0.01) (Figure 6C).

### 3.7. Verification of DASs and DAPAs

Three DASs and three DAPAs were selected for RT-PCR (Figure 7A) and 3′RACE (Figure 7B) validation, and the result suggested that the number of splicing sites was consistent with the sequencing data, confirming the reliability of the sequencing data used in this current work.

## 4. Discussion

Under stress conditions, cells can selectively splice mRNA or alter polyadenylation sites through various post-transcriptional regulation methods, enabling a single gene to produce multiple distinct protein variants. These adaptations allow the cells to respond to diverse environmental conditions and fulfill various functional requirements. Our current understanding of honeybee transcriptomes is primarily derived from gene expression data obtained through NGS technologies. However, the study of the post-transcriptional regulation mechanisms in honeybees is constrained by the absence of complete and sufficiently long read lengths from NGS data. Here, PacBio sequencing makes it possible to identify the complexity of AS in *A. cerana* at the whole-transcriptome level and to obtain accurate AS and APA genes through the correction of second-generation data. In this work, we re-annotated *A. cerana* using PacBio third-generation sequencing and identified 25,443 AS events and 2821 APA genes in the blank control, 4 °C chilling coma, and chilling coma recovery groups at three age stages (*Ac*3d, *Ac*10d, and *Ac*21d). A large number of AS events and APA genes obtained with PacBio and Illumina technology lay the foundation for analyzing the mechanism of bee cold stress response in post-transcriptional regulation.

The transcriptome of *A. cerana* is complex due to extensive post-transcription regulation. Different types of AS have diverse regulatory effects on biological activities. A3 refers to the fact that, in the process of mRNA splicing, the spliceosome can select different 3′ splice sites, so as to include or exclude specific exons in different transcripts [30]. During the adaptation of *Oncorhynchus mykiss* to salinity, A3 leads to the disruption or even loss of functional RNA recognition motif (RRM) domains in *HNRNPA0*, *HNRNP1A*, *HNRNPLB*, and *HNRNPC* genes, hindering the interaction between *HNRNP* genes and pre-mRNA. This, in turn, affects the splicing mode and mRNA stability of downstream target genes, thereby regulating the high osmotic pressure produced by fish salt adaptation [31]. The adaptation of Tibetan pigs to a hypoxic environment is also regulated by the gene expression level of A3, which has many key immune regulation and anti-inflammatory effects on the lungs of pigs under hypoxia [32]. In terms of temperature tolerance, A3 has been reported to be involved in the regulation of cold tolerance genes. The *MTJMJC5* gene of Medicago sativa can produce four types of variable shear isoforms. When exposed to cold treatment, the variants that possess a shared 3′ AS site within the second intron exhibited significant up-regulation, whereas the authentic protein-encoding variant and the variant containing a premature termination codon, which only undergo 3′ AS at the first intron, were down-regulated [33]. The alternative first exon refers to the process in which the spliceosome can select different first exons during mRNA splicing, thereby including different exons in different transcripts. This AS mechanism plays an important role in gene expression, allowing a single gene to produce a variety of different mRNA transcripts and protein variants. The transcription co-activator *PGC-1 α* of mice is involved in a variety of metabolic regulation processes during cold exposure. The AF subtype of *PGC-1 α* is up-regulated in brown adipose tissue in response to cold exposure and is involved in the process of energy consumption and heat production [34]. The AF subtype of human *PGC-1 α* has the same function as that of the same found in mice, and both participate in the response to cold stress by regulating energy metabolism [35]. In this study, we detected that A3 and AF were the two types of variable shear events with the highest frequency. This is different from the proportion of bees as types in Changbai Mountain after overwintering. We speculate that A3 and AL are mainly involved in the short-term cold stress response of bees, and the regulatory effect of cold acclimation caused by long-term cold stress is not significant. Bees need different strategies to deal with the environmental challenges on these two different time scales [36].

In this study we detected several hundred differential splicing events in the cold stress response system of honeybees. In the process of cold stress, *IDH3G*, *GPDH*, *HSP70*, *RPS6KA3*, and *SAMUI* encoded energy metabolism, and oxidative-stress-related proteins represented up-regulation at 3 days old. In the process of cold tolerance, *IDH3G*, *GPDH*, *HSP70,* and *SAMUI* represented down-regulation at 10 days old. It is particularly noteworthy that *HSP70*, coding gene *HSP70*, showed a significant and sustained upward trend in the process of CK-T-TR, and the expression trend of its ubiquitination synergy *BAG* was highly consistent with that of *HSP70*. *HSP70* is a kind of protein expressed under the condition of cell stress, which mainly includes the following two highly conserved regions: the N-terminal ATPase domain and the C-terminal peptide binding domain. The C-terminus of the BAG protein contains a conserved region unique to the protein family BAG domain. *HSP70* responds to different types of stressors and plays a role in maintaining cell function, gene regulation, and protein transport under stress conditions [37]. In honeybees, *HSP70* plays a role in heat stress [38], cold stress [39], and drought stress [40]. HSP70 participates in winter diapause of *Sitodiplosis mosellana* [41]. Our study showed that the high expression of *HSP70* and *BAG* in the process of cold stress was accompanied by the complexity of variable shear events. RI in the HSP70 protein only played a role in the tr group. These differential shears may be the post-transcriptional regulation pathway of *HSP70* expression in response to cold stress in *A. cerana*.

APA is an RNA-processing mechanism that generates different 3′ ends on mRNA and other RNA polymerase II transcripts. APA is widely present in all eukaryotes and considered a major mechanism of gene regulation. APA exhibits tissue specificity and plays an important role in cellular proliferation and differentiation. Due to the presence of cis elements involved in various aspects of mRNA metabolism within the 3′ UTR, 3′ UTR-APA can significantly impact post-transcriptional gene regulation through the modulation of mRNA stability, translation, nuclear export, and cellular localization [42]. The two core cis elements in the cleavage and polyadenylation of RNA are protein recognition sites [43]. The first core cis element is called the polyadenylation site (PAS), which is a highly conserved AAUAAA hexamer motif, 10–35 bp upstream of the 5′ end of the cleavage site, and the other is a variable sequence rich in U/UG, 15–30 bp downstream of the 3′ end of the cleavage site. In the process of polyadenylation, cleavage- and polyadenylation-specific factors (CPSF) are combined with protein through PAS, and the variable sequence rich in U/UG is directly combined with cleavage stimulating factor (CSTF) to determine the cleavage site and finally complete the cleavage and polyadenylation of the 3 ‘end of mRNA [44,45]. This process is crucial for the maturation and stability of mRNA. The first insect used to study the variable polyadenylation sites was *Drosophila melanogaster*, and the main motif was AAUAAA [46]. At present, 2821 genes were regulated by APA in our research. AAUAAA, AUAAA, and UUUUUCUU were highly conserved polyadenylation signals among all of the polyadenylation signals identified by us. The results showed that the distribution of PAS variants was consistent with that of fly, human [45], and *Spodoptera frugiperda*, indicating that the polyadenylation process was relatively conservative in different animals [47]. The PAS frequency of AUAAA/AUAAA is much higher than that of other PAS variants, and a large number of loci are used for CPSF recognition. *UGCG* and *CUBN* in the CK-T-TR group showed the same trend as that of the main gene expression, being first down-regulated and then up-regulated. These results suggest that APA may participate in the cold stress response of *A. cerana* by regulating the transport of energy substances.

Worker bees with different functions have different sensitivities to ambient temperature, which affects their behavior, physiological response, and adaptability to temperature changes. These differences are crucial for the survival and reproduction of bees. Research indicates that during chill stress, the older worker bees mainly transfer heat to the nest through the chest, while the younger worker bees mainly receive heat from the surrounding nest [48]. The response mechanisms of different functional worker bees to low-temperature stress are different. Our study showed that the proportion of poly(A) sites exceeding five in APA of *Ac*21d worker bees was significantly higher than that of *Ac*3d and *Ac*10d worker bees. The post-transcriptional regulatory events in the *Ac*21d worker bees under temperature stress exhibited greater complexity and diversity, possibly linked to the variable environment encountered during frequent foraging outside of the nest. Especially in DAPAs, the expression of *CUBN* in CK-T-TR increased first and then decreased in *Ac*21d, contrasting with the trends observed in *Ac*3d and *Ac*10d worker bees, with poly(A) site changes occurring exclusively in *Ac*21d worker bees. Cubilin, a large endocytic receptor with a molecular weight of 460 kDa, binds to various ligands and is involved in vitamin B12 absorption in the intestine, apolipoprotein A-I catabolism in the proximal tubules, and protein reabsorption in the kidney [49]. Recent studies have shown that Cubilin and its homologous protein Amnionless regulate proteostasis in muscle and brain tissues in fruit flies by mediating protein reabsorption (CAMPR) in renal cells, thereby affecting lifespan and health [50]. These results indicate that cold environments may have a negative impact on the lifespan of bees, especially in the development stage of bees, but it may also lead to the prolongation of the lifespan of bees in some cases (such as short-term cold stimulation in forger bees), which may be mediated by APA. The outcomes of this research can offer fresh perspectives on post-transcriptional control mechanisms, such as AS and APA, in response to cold stress in honeybees. Additionally, our findings foster further investigation into the cryobiology of insects.

## 5. Conclusions

In summary, post-transcriptional regulation through AS and APA in *A. cerana* may respond to chilling stress by regulating various metabolic pathways and antifreeze proteins, offering valuable insights for insect cold resistance research and molecular breeding of cold-resistant bee strains (Figure 8).

## Figures and Tables

**Figure 1 insects-15-01006-f001:**
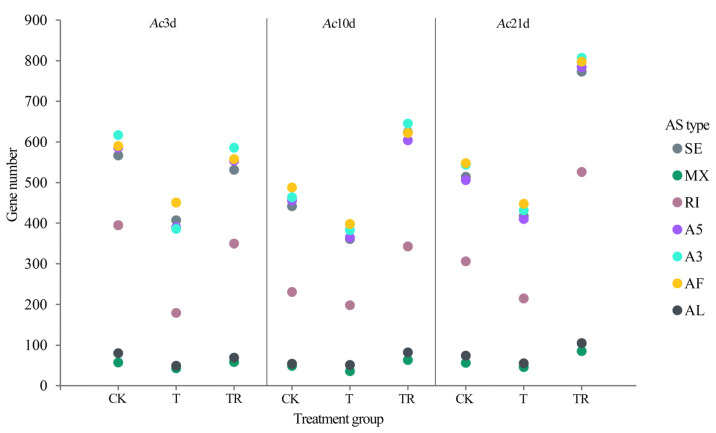
The type of AS events.

**Figure 2 insects-15-01006-f002:**
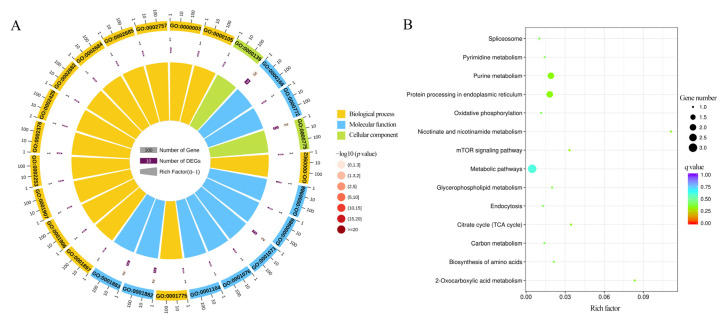
GO classification (**A**) and KEGG pathway annotation (**B**) by DASs.

**Figure 3 insects-15-01006-f003:**
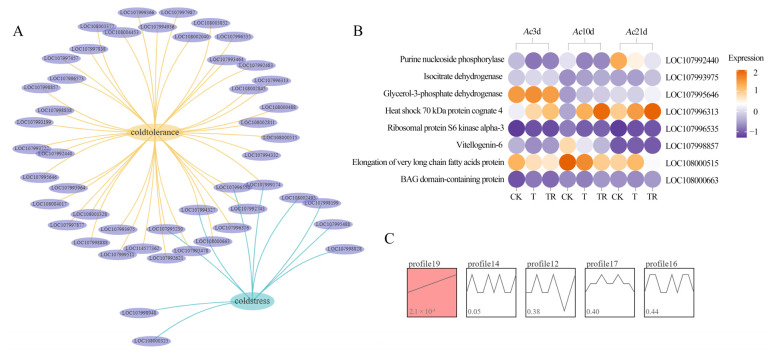
Analysis of chilling stress response factor-associated DASs. (**A**) DiVenn analysis of cold stress (T vs. CK) and cold tolerance processes (TR vs. CK). The large node in the figure is the group name, and the small node is the gene ID. (**B**) The cluster expression analysis of chilling stress response factor-associated DASs. (**C**) The trend analysis of DASs.

**Figure 4 insects-15-01006-f004:**
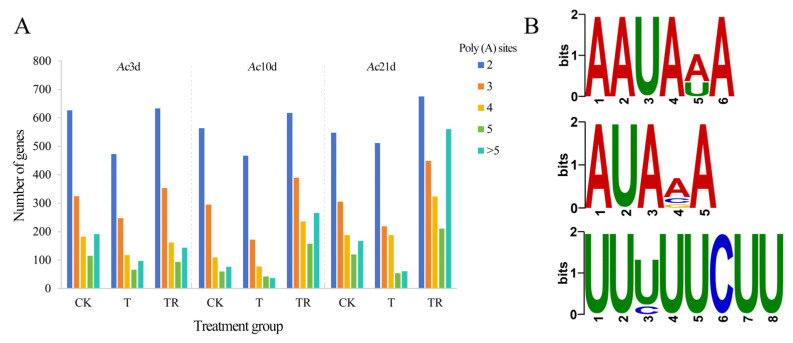
Identification of APA. (**A**) Number statistics of full-length transcripts containing various numbers of APA sites; (**B**) Motifs identified at 50 nt upstream of APA sites.

**Figure 5 insects-15-01006-f005:**
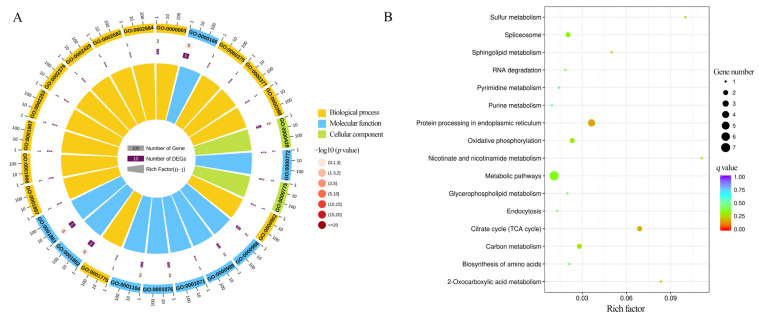
GO classification (**A**) and KEGG pathway annotation (**B**) by DAPAs.

**Figure 6 insects-15-01006-f006:**
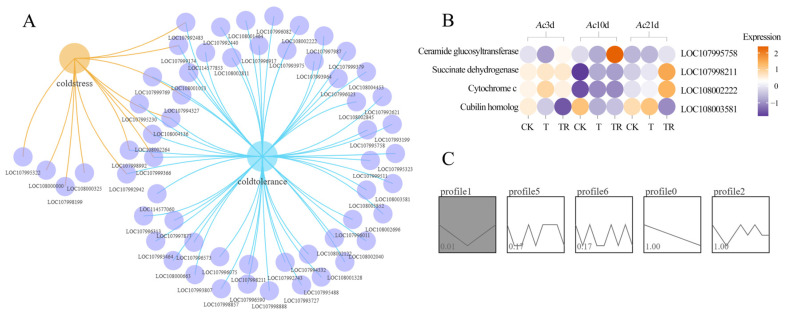
Analysis of chilling stress response factor-associated DAPAs. (**A**) DiVenn analysis of cold stress (T vs. CK) and cold tolerance processes (TR vs. CK). The large node in the figure is the group name, and the small node is the gene ID. (**B**) The cluster expression analysis of chilling stress response factor-associated DAPAs. (**C**) The trend analysis of DAPAs.

**Figure 7 insects-15-01006-f007:**
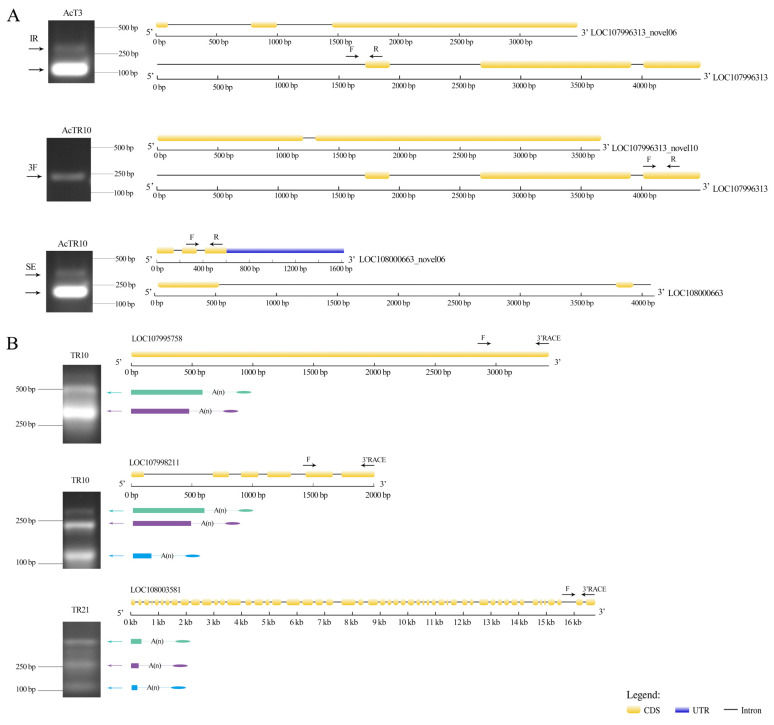
PCR verification of AS events (**A**) and APA sites (**B**).

**Figure 8 insects-15-01006-f008:**
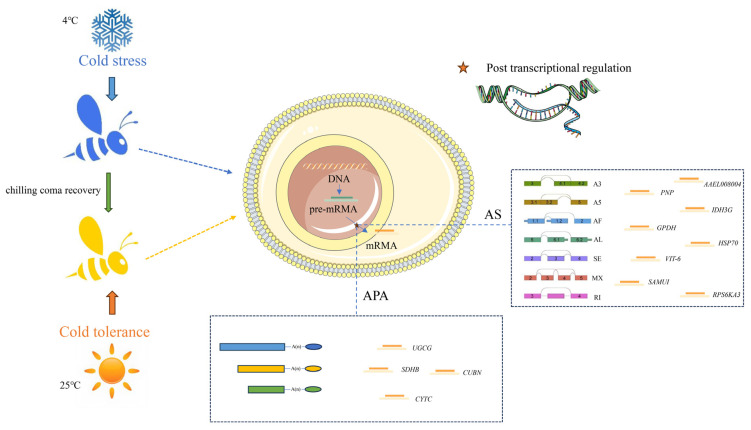
*A. cerana* activate post-transcriptional regulation of AS and APA after cold stress. *PNP*, *IDH3G*, *GPDH*, *HSP70*, *RPS6KA3*, *VIT-6*, *AAEL008004*, and *SAMUI* responded to cold stress through AS, while *UGCG*, *SDHB*, *CYTC*, and *CUBN* responded to cold stress through APA.

## Data Availability

Sequence data that support the findings of this study have been deposited in the National Center for Biotechnology Information with the BioProject accession code gca_029169275.1.

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
