# Peer review of "Alternative Splicing and Alternative Polyadenylation-Regulated Cold Stress Response of Apis cerana"

_insects, 2024, doi:10.3390/insects15121006_

Round 1

Reviewer 1 Report

Comments and Suggestions for Authors

This study used PacBio and Illumina sequencing platform to obtain the full-length transcriptome data of Apis cerana workers. They identified the alternative splicing and alternative polyadenylation and compared the difference between workers under cold stress and the control in different age group. They also verified the isoforms of three differentially expressed alternative splicing and differentially expressed alternative polyadenylation sites. The study revealed that the post-transcriptional regulation of different metabolic pathways and antifreeze proteins may respond to chilling stress in A. cerana. It would be helpful for our understanding of the molecular mechanisms of cold resistance in A. cerana. However, there are several improvements need to be made.

Major points:

1.     The improvements of expression are necessary to make it easier to be understood. Here are just a few examples: In the abstract, you should make it clear what does “Ac3d, Ac10d and Ac21d” mean. Please check and make the sentence “The results revealed that DASs primarily involve the regulation of high such as HSP70 and BAG, with the expression levels of these genes showing an upward trend” complete. In the introduction, the sentences “The low temperature test brought by the long winter is the premise of its population survival and reproduction” and “At present, the cold tolerance of insects is mostly concentrated in solitary insects, and the individual temperature regulation of social insects is relatively less” need to be revised to make it more clear.

2.     Some statements are repetitive. For example, in the first and the last paragraph of the introduction, the information about A. cerana is repeated.

3.    For the materials and methods: how many workers were sequenced for each age group of CK, T and TR? How many sequencing duplications for each group?

4.     Line 164: what is SE, MX, A5, A3, AF, AL and IR short for?

5.     Line 239: what is collected bees? Do you mean foragers?

6.     Line 417: Drosophila melanogaster should be italics.

7.     The figures in the paper needs to be improved to make them with higher resolution.

8.     In the result and the discussion, you mainly mentioned the total AS events and APAs identified among all groups (between T vs CK and TR vs CK). There is very little analysis of the differences between the two processes (cold tolerance and cold stress). In addition, little is mentioned about how they are related to the age groups.

Comments on the Quality of English Language

The English could be improved to more clearly express the research.

Author Response

Comments1: The improvements of expression are necessary to make it easier to be understood. Here are just a few examples: In the abstract, you should make it clear what does “Ac3d, Ac10d and Ac21d” mean. Please check and make the sentence “The results revealed that DASs primarily involve the regulation of high such as HSP70 and BAG, with the expression levels of these genes showing an upward trend” complete. In the introduction, the sentences “The low temperature test brought by the long winter is the premise of its population survival and reproduction” and “At present, the cold tolerance of insects is mostly concentrated in solitary insects, and the individual temperature regulation of social insects is relatively less” need to be revised to make it more clear.

Response: Thank you for your constructive feedback. In order to improve the clarity and comprehensibility of the manuscript, we have carefully considered your suggestions, modified the full text, and marked the modified sentences in red.

Comments2: Some statements are repetitive. For example, in the first and the last paragraph of the introduction, the information about A. cerana is repeated.

Response: Thank you for pointing out the redundancy in our manuscript. We have identified and removed the repetitive statements regarding Apis cerana from the last paragraphs.

Comments3: For the materials and methods: how many workers were sequenced for each age group of CK, T and TR? How many sequencing duplications for each group?

Response: Following your kind comment, we added related description to give detailed information for each group.

Comments4:  Line 164: what is SE, MX, A5, A3, AF, AL and IR short for?

Response: Thank you for your inquiry regarding the abbreviations used in our manuscript. These abbreviations have full names in the introduction section Line79-81 of the paper.

Comments5: Line 239: what is collected bees? Do you mean foragers?

Response: Thank you for pointing out the wrong use of professional terms. We made corresponding correction in the revised manuscript.

Comments6: Line 417: Drosophila melanogaster should be italics.

Response: Following your kind comment, we made thorough check throughout the whole manuscript and necessary correction.

Comments7: The figures in the paper needs to be improved to make them with higher resolution.

Response: Thank you for your feedback regarding the quality of the figures in our manuscript. We have re-uploaded high-quality images.

Comments8: In the result and the discussion, you mainly mentioned the total AS events and APAs identified among all groups (between T vs CK and TR vs CK). There is very little analysis of the differences between the two processes (cold tolerance and cold stress). In addition, little is mentioned about how they are related to the age groups.

Response: Thank you for your observation. We have taken your comments into account and have expanded our analysis to more thoroughly explore the differences between cold tolerance and cold stress processes, as well as their relationships with the different age groups. This additional analysis provides a more nuanced understanding of the post-transcriptional regulation in Apis cerana in response to cold stress.

Reviewer 2 Report

Comments and Suggestions for Authors

Please add author name of the Aphis cerena in the title.

In scientific writing rules, species names are written binomially. While genus names can be abbreviated, species names cannot be abbreviated. The writing of subspecies is called trinominal. After the species name, 'ssp.' is added, and the subspecies name is written. 'ssp.' is written in normal text, not italicized. In any case, the species name should not be abbreviated. Please correct throughout the text.

Scientific names should be written in full form at the beginning of a paragraph and sentence.

Author Response

Comments1: Please add author name of the Aphis cerena in the title.

In scientific writing rules, species names are written binomially. While genus names can be abbreviated, species names cannot be abbreviated. The writing of subspecies is called trinominal. After the species name, 'ssp.' is added, and the subspecies name is written. 'ssp.' is written in normal text, not italicized. In any case, the species name should not be abbreviated. Please correct throughout the text.

Scientific names should be written in full form at the beginning of a paragraph and sentence.

Response: Thanks for your kind comment. We have modified the scientific names of species in the full text in accordance with the naming rules, and marked the modified places in red.

Reviewer 3 Report

Comments and Suggestions for Authors

The manuscript "Alternative Splicing and Alternative Polyadenylation regulated cold stress response of Apis cerana cerana". The topic is interesting enough to investigate and the experiments are well-designed. The effects of low temperature on the alternative splicing and alternative polyadenylation in Apis cerana cerana were evaluated, the result has practical significance for the development of molecular breeding targets for cold-resistant strains of bees. This is a very well-written manuscript; my concern is that there is some important information missing in the Materials and Methods section

1.       The “Simple Summary” should be supplied before the “Abstract”.

2.       Line 18, the “the behavioral strategies and physiological mechanisms of honeybee cold resistance” should be “honeybee cold resistance's behavioral strategies and physiological mechanisms”.

3.       Line 22, the “honeybee” should be “honeybees”.

4.       Line 54, “The” should be added before “honeybee”.

5.       Line 73, “including” should be “include”.

6.       Line 82, “the response of environment stress” should be “the response to environmental stress”.

7.       Line 92, “on the basis of” should be “based on”.

8.       Line 115, First appearance of NGS, and the full name of which added

9.       Line 124-126, To add a few case studies or literature on alternative splicing and alternative polyadenylation in other insects using third-generation sequencing technology.

10.    Line 140, “Cold stress response in honeybee.” should be “cold stress response in honeybees”

11.    Line 143-144, How many bee colonies were used in this study? What was the method of collection of these bees of different ages? And theses information also should be introduced.

12.    Line 145-148, How many replicates per treatment group? How many worker bees per replicate? Why did you choose 4 degrees Celsius? Is this consistent with actual production? Did the survival rate of these worker bees decrease after the low-temperature treatment?

13.    Line 151, What is the number of worker bees used for RNA extraction per treatment? And add the number of replicates and the number of samples per replicate

14.    Line 199,A. c. cerana” should be italicized.

15.    Line 362, Why these “Three DASs and three DAPAs” were selected?

16.    Line 382, “Cold stress” should be “cold stress”

17.    Line 457, the style of reference should be carefully modified following the Instructions for Authors.

Author Response

Dear Reviewers,

We appreciate your comments and suggestions of great importance, which significantly improve the quality of our work and manuscript. Accordingly, we seriously checked and modified the manuscript, and all revision were showed in red in the revised version of manuscript. Point-to-point response to review comments were as follows:

comments1: The “Simple Summary” should be supplied before the “Abstract”.

Response: Thank you for your suggestion. We have added a "Simple Summary" section prior to the "Abstract" to provide a brief overview of our study.

comments 2: Line 18, the “the behavioral strategies and physiological mechanisms of honeybee cold resistance” should be “honeybee cold resistance's behavioral strategies and physiological mechanisms”.

Response: Thank you for your valuable recommendation, based on which we improved the Abstract.

Comments3: Line 22, the “honeybee” should be “honeybees”.

Response: Corresponding correction was made following your comment.

Comments4: Line 54, “The” should be added before “honeybee”.

Response: Corresponding correction was made following your comment.

Comments5: Line 73, “including” should be “include”.

Response: Corresponding correction was made following your comment.

Comments6: Line 82, “the response of environment stress” should be “the response to environmental stress”.

Response: Corresponding correction was made following your comment.

Comments7: Line 92, “on the basis of” should be “based on”.

Response: Corresponding correction was made following your comment.

Comments8: Line 115, First appearance of NGS, and the full name of which added

Response: Thank you for your attention to detail. We have added the full name of "Next-Generation Sequencing" (NGS) on its first appearance in the manuscript.

Comments9: Line 124-126, To add a few case studies or literature on alternative splicing and alternative polyadenylation in other insects using third-generation sequencing technology.

Response: Thank you for your insightful suggestion. We have now included additional case studies and literature references on alternative splicing and alternative polyadenylation in other insects, specifically utilizing third-generation sequencing technology.

Comments10: Line 140, “Cold stress response in honeybee.” should be “cold stress response in honeybees”

Response: Corresponding correction was made following your comment.

Comments11: Line 143-144, How many bee colonies were used in this study? What was the method of collection of these bees of different ages? And theses information also should be introduced.

Response: Thanks for your suggestion. We have revised the manuscript to include these details in the Materials and Methods section to provide a clearer understanding of our sampling procedures.

Comments12: Line 145-148, How many replicates per treatment group? How many worker bees per replicate? Why did you choose 4 degrees Celsius? Is this consistent with actual production? Did the survival rate of these worker bees decrease after the low-temperature treatment?

Response: Thank you for your inquiry. We will reply to your questions point by point.

(1) Number of Replicates per Treatment Group: We conducted our experiments with three replicates per treatment group to ensure the reliability and reproducibility of our results.

(2) Number of Worker Bees per Replicate: We used 30 worker bees per replicate.

(3) The rationale behind selecting 4°C as the temperature treatment is grounded in the climatic data specific to the region where our study was conducted. December, which is the primary month for honey collection in Guizhou, has an average daily low temperature of 4°C. This temperature is representative of the natural conditions that honey bees in this region are exposed to during the winter season, particularly when they are foraging and when the capped brood stage is critical for their development. By choosing 4°C, our study aims to simulate and investigate the physiological responses of honey bees to the actual environmental stress they face in their natural habitat during the winter months. This temperature is consistent with the real-world challenges that the bees encounter, which is crucial for understanding their resilience and survival strategies under such conditions.

(4) We subjected bees to cold treatment at 4 ℃ for 12 hours, and the survival rate was 100%. We recorded the cold coma recovery time of honeybees after freezing at 4 ℃ for 0 h, 4 h, 6 h, 8 h, 12 h, and 14 h. When the freezing time was 6 h, the cold coma recovery time was significantly longer than that of 0 h and 4 h, but not significantly different from that of 8 h, 12 h, and 14 h (Figure 1). Therefore, in this experiment, 4 ℃ is the treatment temperature and 6 hours is the treatment time.

Figure 1 Summary of chill coma recovery time in Apis cerana

We hope this provides the necessary clarification and context for our experimental design and findings.

Comments13: Line 151, What is the number of worker bees used for RNA extraction per treatment? And add the number of replicates and the number of samples per replicate

Response: Thank you for your attention to detail. We have added this description in Materials and Methods. “For the Illumina RNA-seq second-generation sequencing, 10 bees were used per sample with three biological replicates. For the PacBio Sequel third-generation sequencing, 15 bees were used per sample.”

Comments14: Line 199, “A. c. cerana” should be italicized.

Response: Corresponding correction was made following your comment.

Comments15: Line 362, Why these “Three DASs and three DAPAs” were selected?

Response: Thank you for your insightful comments and questions regarding our selection of genes for validation. The genes in question were selected from a pool of eight key regulatory DASs and four DAPAs. We conducted a sequence alignment of the alternatively spliced isoforms of the key regulatory DAS genes and analyzed the upstream and downstream sequence information surrounding the splice sites. Based on this analysis, we identified two isoforms (with a size difference in the spliced segment of >100bp) that could be distinctly differentiated by gel electrophoresis bands, and designed specific primers accordingly. Prior to the formal experiments, we performed preliminary experiments, in which these genes exhibited significant differences, thereby qualifying them as subjects for validation. We hope that this detailed explanation addresses your concerns and provides a clear rationale for our experimental design.

Comments16: Line 382, “Cold stress” should be “cold stress”

Response: Corresponding correction was made following your comment.

Comments17: Line 457, the style of reference should be carefully modified following the Instructions for Authors.

Response: Thank you for your guidance. We have carefully reviewed and updated the reference style to comply with the journal's Instructions for Authors. The references have been formatted accordingly to ensure consistency and adherence to the publication's standards.

Reviewer 4 Report

Comments and Suggestions for Authors

The full-length transcriptome data described in this manuscript is awesome and it has definitely great importance in elucidating cold tolerance of bees based on post-transcriptional regulation. However, I regret that the conclusion described in the manuscript is somewhat unsatisfactory compared with the potential utility of the data. Especially, it is disappointing that the identified differential alternative splicing is only 48. I could not exactly understand the criteria of the authors to judge “differential”. I think that most critical differences in alternative splicing events among different day-old and temperature condition workers are overlooked. I hope that the authors could show more novel findings in another opportunity. I only point out several revisions to be improved as follows.

Lines 79-81: Explanation of abbreviations of AS subtypes only appeared here. It is quite troublesome to refer back every time when such uncommon abbreviations are appeared. Such explanation should be moved to Results, at least.

 Lines 157-161: Such strings of numbers are nearly impossible to understand, thus it should be organized in a table.

Figure 1: This figure lacks legends and hard to understand. It should be shown in a table.

 Figure 2: This figure also lacks legends and hard to understand. I could not understand what means each color of squares in panel B of Figures 2, 3, 5 and 6 at all.

 Line 378: Al should be written as AL.

Author Response

comments1: Lines 79-81: Explanation of abbreviations of AS subtypes only appeared here. It is quite troublesome to refer back every time when such uncommon abbreviations are appeared. Such explanation should be moved to Results, at least.

Response: Thank you for your constructive feedback on the clarity of our manuscript, particularly regarding the abbreviations used for alternative splicing (AS) subtypes. We added the full names of these abbreviations in the materials and methods section.

Comments2: Lines 157-161: Such strings of numbers are nearly impossible to understand, thus it should be organized in a table.

Response: Thank you for your valuable comments on the presentation of data in our manuscript. We agree that the long string of figures is difficult to understand, and have adopted your suggestion to sort the relevant data into a table form.

Comments31: Figure 1: This figure lacks legends and hard to understand. It should be shown in a table.

Response: Thank you for your feedback on Figure 1. We added the missing legends in the picture and added the data in table to the appendix.

Comments4: Figure 2: This figure also lacks legends and hard to understand. I could not understand what means each color of squares in panel B of Figures 2, 3, 5 and 6 at all.

Response: Thanks for your attention to detail. We added the missing legends in the picture.

Comments5: Line 378: Al should be written as AL.

Response: Corresponding correction was made following your comment.
